# The Design of WASPEC: A Fully Personalised Moodle System Using Semantic Web Technologies

Ufuoma Chima Apoki 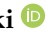

Faculty of Computer Science, Alexandru Ioan Cuza University, 700506 Iasi, Romania; ufuomaapoki@gmail.com

**Abstract:** Personalisation in e-learning systems has become a major research area in recent times, as online learning is gradually evolving to become a major part of formal education. While there exist several learning management systems with a wide range of capabilities, one thing that remains inefficient is a standard framework for sharing knowledge across different platforms and, also, the inability of such systems to provide personalisation to the learning process by default. A large number of systems that have been implemented to provide personalisation apply few parameters and are course-specific; thus, flexibility, reusability, and scalability are greatly reduced. In this paper, we propose a framework for personalised learning, Weighted Agent System for Personalised E-learning Curriculum (WASPEC) implemented with Moodle, which is independent of the learning management system and provides the possibility of incorporating multiple personalisation parameters. This is accomplished with the combined use of web services, semantic web ontologies, and pedagogical agents, providing dynamic personalisation in the background of the e-learning system. This also provides added advantages of the possibility of sharing knowledge with other systems and reusability

**Keywords:** learning management systems; personalised learning; ontologies; semantic web; web services; pedagogical agents; personalisation parameters

## 1. Introduction

Initially, e-learning systems were intended to provide learners with information. However, beginning in the early 1970s, they became more interactive [1]. E-learning became popular in the late twentieth century due to rapid advances in computing power and the development of the internet. In recent years and in light of situations such as the COVID-19 pandemic, e-learning has evolved from an optional mode of instruction to a required component of formal education.

As e-learning systems evolved to be more interactive, a variety of applications were created to accommodate both the interactive and explorative aspects of the learning process on the web. Examples of such include virtual learning environments (VLE) which allow course instructors to share educational materials and communicate with learners over the web, learning management systems (LMS), which focus more on management in the delivery of learning content to students, allowing for learner participation and progress to be tracked, and content management systems (CMS), which are data repositories where any kind of data (sounds, documents, videos, pictures) can be stored.

### 1.1. Personalisation in E-Learning

According to the National Academy of Engineering, one major concern, challenge, and interest of the 21st century in web-based course delivery is the concept of personalised learning experiences, which account for different learner characteristics that have the potential to influence learning [2]. To design personalized e-learning environments, researchers and pedagogues either build customised e-learning platforms (which are usually restricted to specific domains) or add extra functionalities to existing platforms such as LMSs, which, by default, do not incorporate adaptivity or personalisation.

Modern technologies that support personalisation in such environments can be grouped into the following categories, as described by [3]:

- Adaptive presentation: The purpose of this is to alter the presentation of content to suit the preferences of learners.
- Adaptive navigation: This involves learners being provided with different learning paths (mainly by means of hyperlinks) through the learning content or the learning platform to achieve individual satisfactory learning.
- Curriculum sequencing: The aim here is to provide the right sequence of concepts or learning resources that each learner prefers or which matches the learner's abilities.
- Intelligent solution analysis: This provides intelligent analysis of assessment items submitted by learners as opposed to yes/no grading.
- Problem-solving support: For this approach, step-by-step insightful assistance is provided to a learner during a session where the learner has to solve a problem, based on the learner's unique needs.

While adaptive presentation and adaptive navigation are mainly integrated into adaptive hypermedia systems, curriculum sequencing, intelligent solution analysis, and problem-solving support are mainly categorised as intelligent tutoring. While there is no clear distinction between adaptive and intelligent learning environments, one distinction is that adaptive systems behave differently based on user preferences and needs, whereas intelligent systems provide the same level of intelligent tutoring to all users [4]. In recent times, however, these approaches overlap when designing personalised web-based learning environments.

Personalisation by extending existing LMSs is usually achieved through curriculum sequencing, where different learning resources (which instruct different concepts or achieve different competencies) can be presented to learners, and adaptive navigation, where links and hyperlinks are adapted to student preferences and features.

### 1.2. Personalisation Parameters

In recent times, there has been a lot of research into various parameters for personalizing learning experiences according to learning preferences and abilities. These parameters can be categorised into the following groups according to [5]: learning preferences, learning status and history, features of the medium of learning, and pedagogical/domain features. Learning preferences include choices a learner can make when provided a list of options. A learner's status and history describe various states of the learner before, during, and after learning sessions. The features of the device, the learner uses at any given particular time, can be used as parameters for personalisation. The structure and relationships in a knowledge domain can also be used for personalisation, as well as different pedagogical approaches.

Parameters are defined by dimensions, which are essentially divergent groupings for which certain characteristics differ. A learner's level of knowledge, for instance, as a personalisation parameter can have dimensions of beginner and advanced. A personalisation strategy is the combination of different dimensions of more than one personalisation parameter.

### 1.3. Standardisation and the Semantic Web in E-Learning

Standardisation has become the main theme in e-learning, intending to develop features of a learning domain and users which can be widely adopted and interchangeable [6]. This promotes interoperability and compatibility between different e-learning platforms. With standardisation, specifications and metadata are described by international organisations such as IEEE and Instructional Management System Global Consortium for e-learning platforms [5,6].

Web 3.0, which defines semantic web technologies, has also become increasingly vital in providing personalisation in LMS [6]. The technologies include RDF (Resource

Description Framework), XML (Extensible Markup Language), and ontologies. These technologies provide a framework for interconnecting and describing objects in a network.

Ontologies, which can be expressed as RDF, are specifications of concepts and relationships between these concepts in a particular domain; and they provide a way of sharing and formalizing knowledge across multiple systems [7]. When ontologies are expressed properly, they can serve as powerful tools for content reuse, inference of new knowledge, deep insights, and the uncovering of hidden relationships in networks. The formalisation and interconnections are particularly useful in making inferences and connections between learning resources and information describing a learner from his/her interactions with the system or his/her preferences.

### 1.4. Pedagogical Agents

Pedagogical agents are software agents that aim to improve a learning environment with their characteristic attributes of longevity, semi-autonomy, proactivity, and adaptivity. Software agents have been widely used since the 1970s in learning environments such as Intelligent Tutoring Systems (ITS) and virtual learning environments (VLE) [8]. Over the years and with substantial advances in interface technology, pedagogical agents have been claimed to generate realistic simulations (as tutors, assistants, co-learners), encourage student engagement, motivation, and responsibility, and generally improve learning experiences and performances by addressing learners' personal and sociocultural needs [8]. An interesting application of pedagogical agents is in a Multi-Agent System (MAS) where intelligent agents collectively cooperate via defined communication protocols to achieve predefined goals, with each agent having specified control in certain activities [9].

### 1.5. Motivation and Objectives of the Paper

It has been observed from multiple surveys in the literature that most e-learning systems accommodate few learner preferences and characteristics in the personalisation of learning [5,10]. While this is a feasible approach because the number of parameters that can be integrated into a single strategy for personalisation is limited, researchers have expressed the need to integrate new personalisation strategies that accommodate different learning characteristics.

In this context, this paper aims to address the following questions:

1. How can learners be modeled with a set of multiple personalisation parameters?
2. How can standardisation and semantic web technologies facilitate personalisation and interoperability within existing learning management systems?

To answer the questions posed above, this paper presents the design and implementation of WASPEC (Weighted Agent System for Personalised E-learning Curricula). This work is built on the model designed in [11] and the approaches towards multi-personalisation in [10]. Hence, an e-learning system is presented with the following features and objectives:

1. Defining a set of metrics that select the most relevant parameters for personalisation for each course based on available learning resources,
2. Implementing a semantic framework that models a domain, maps the features of learning resources to personalisation parameters, and uses the mappings to recommend learning resources according to learners' preferences,
3. Combining these personalisation technologies with an LMS (Moodle, specifically) to provide personalisation to learners.

The rest of the paper is structured as follows: Section 2 presents related works specific to this paper. Section 3 presents a conceptual framework, describing the components and architecture of WASPEC in detail. In Section 4, the procedure of implementation is explained. Section 5 presents the process and tools of evaluating the system, while Section 6 presents the results. Section 7 compares the WASPEC learning platform to similar models of e-learning platforms that provide personalisation by extending LMSs. The paper ends with conclusions and future work in Section 8.

## 2. Literature Review

In recent years, there has been an increase in the demand for e-learning systems that provide different learning paths based on the diverse preferences of learners. In general, the literature on systems that provide some level of personalisation in various aspects of e-learning is virtually limitless. Designs that incorporate personalisation within existing LMSs, with a particular emphasis on standardisation approaches, semantic web technologies, and multiple parameters, are featured in this section.

### 2.1. Search Strategy

To answer the questions posed in Section 1.5 required a thorough search of personalised learning systems from (but not exclusively limited to) the following online databases: Google Scholar, Science Direct, IEEE Xplore, ResearchGate, SpringerLink, and ACMPortal. The publications and articles included review articles, empirical systems, and theoretical models from conference proceedings, journals, book sections, and reports from international organisations. The search items included: 'adaptive learning', 'personalised e-learning', 'adaptive educational systems', 'adaptive hypermedia systems', 'metadata adaptation', 'pedagogical agents', 'semantic web technologies', 'ontologies', 'adaptive learning objects', 'personalisation parameters'. A complementary search strategy included reference lists of papers that were considered important to this research.

### 2.2. Inclusion and Exclusion Criteria

The following were used as guidelines for the selection of core articles or reports:

- Research papers that included empirical models with multiple parameters in the personalisation process,
- Papers that provided personalisation by incorporating technologies (such as plugins or web services) with a learning management system (LMS),
- Papers that achieved personalisation with metadata standardisation, ontologies, or pedagogical agents,
- Papers that achieved personalisation with metadata standardisation, ontologies, and pedagogical agents.

Exclusion criteria included papers that did not have a conclusive implementation approach and publications in a different language than English.

### 2.3. Related Work

The authors in [12] proposed and implemented an approach to provide personalisation in a Moodle platform using two popular models of learning styles VAK (Visual, Auditory, and Kinesthetic) model and the global/sequential class of the FSLS model. The approach involved creating learning resources for all divergent dimensions represented, which meant the course instructor had to develop six different learning materials for assignments, quizzes, and learning resources.

A platform, MAL (Moodle Adaptive Learning), was designed in [13] that provided personalisation to learners based on their learning styles. Semantic web technologies, which run in the background, were specified and implemented to support personalisation. Specifically, an ontology, MAU, was developed to define the relationships between learners, learning resources, and the activity of learners on the platform.

An approach for personalizing learning resources on Moodle is proposed in [14] based on the FSLS model. This approach provides group personalisation using data mining techniques and clustering. The implementation was to make the personalisation process time-efficient and economically justified. The process follows learners having to go through an introductory course and also respond to questionnaires. Data exploration and clustering of learners into different groups, based on their learning styles and characteristics, is carried out from data retrieved from the first phase. Courses are then adapted by providing different learning experiences based on the teaching materials, examination, and learning activities.

In [15], the authors propose a system, Personalized Learning Management System (PLeMSys) which provides personalisation on Moodle through plugins according to the level of knowledge of learners and their learning style. Two levels of personalisation were specified in this model: the first level of personalisation according to the level of knowledge of the student, and the second level according to their learning styles. For learning styles, personalisation is executed by varying learning resources based on multimedia formats, theoretical, and practical content. The approach is also dependent on data collection and processing of learning activities and results from the learning style questionnaires, after which patterns are determined through clustering and association rules. Personalisation is then carried out based on these analysed data.

The authors in [16] designed personalised learning paths based on a learner's background and learning objectives. Personalisation was provided using AI planning techniques in the Moodle implementation. A PDDL (Planning Domain Definition Language)model was created specifically to map the description of a course (activities and relationships between learning resources) and student features. The implementation and design are quite adaptable and can be integrated with the existing LMS.

A system proposed in [17] enabled course adaptivity based on the learning styles according to FSLS. The experiment is carried out on the Moodle e-learning system by supplementing it with an add-on facility. Because Moodle is static in nature, more user interface components could not be integrated with the application.

A Sharable Auto-Adaptive Learning Object (SALO), designed according to SCORM specification, is presented in [18]. The proposed SALO is intended to be fully autonomous, dynamic in expressing its content, and adaptive in behaviour, making it interchangeable and shareable among different learning management systems. Adaptive presentation and navigation are used to personalise the experience.

## 3. Waspec: Conceptual Framework

In this section, we detail the architecture of WASPEC, the system components, and the framework behind the development of the system. The architecture of WASPEC, which is highly modular, is drawn from similar proposals of personalised LMSs and adaptive e-learning environments supported by semantic technologies [19].

### 3.1. Parameters for Personalisation

To address the issue of integrating multiple parameters, WASPEC's approach towards personalisation is neither a single parameter nor a limited set of parameters, but a set of possible parameters. When personalisation incorporates multiple parameters, several challenges naturally arise. The first is the amount of time it will take for initial testing to place learners in appropriate groups/categories. Secondly, is the time and resources required to put together learning resources that satisfy every personalisation strategy from a combination of a set of parameters. Finally, what procedure will be used to select relevant personalisation parameters for each course.

Selecting the most relevant and appropriate parameters (from the possible set of parameters) for a personalisation strategy greatly reduces the amount of time and resources that would be required to put together learning resources for each course, as suggested in [10]. This also cuts down the time for the process of initial categorisation of learners. Learners only have to go through tests/tasks or submit some information required for relevant parameters of courses they register for. The system saves such information for future use and analysis in personalizing other courses.

The set of possible parameters (and dimensions) that were adopted in this design include the level of knowledge (beginner, intermediate, advanced), language (English, Romanian), learning style (Felder-Silverman Learning Style [20]), media preference (text/image, audio, video, illustration), and motivational level (low, medium, high). The learner's level of knowledge, in a particular knowledge domain, has been observed to be the most dominant personalisation parameter in e-learning systems [4,5,10].

The authors in [10] suggested different approaches that can be used in selecting relevant parameters for personalisation based on the learning resources in a course and/or the choice of a pedagogue/domain expert. Hence, there are two sets of possible parameters for WASPEC. The first set of parameters determines what the learner has (or wants) to learn (the level of knowledge) and the second set determines how the learner goes about learning it (language, learning style, media preference, motivational level).

For learning style, the Felder-Silverman Learning Style (FSLS) model, which is widely used in several designs [21] for personalisation, is used. The argument for the application of learning styles in personalizing educational content is in the following phases [22]:

1.  Learners will demonstrate a preference regarding their 'style' of learning,
2.  Individuals differ in their ability to learn specific types of information,
3.  Better educational outcomes will result from the 'matching' of instructional design to an individual's Learning Style, as defined by one of the aforementioned classifications.

The arguments against learning styles usually center on the third phase of 'matching'. While there is a lot of empirical research on both sides of the argument (for, or against [22,23] learning styles), the approach to the application of learning styles in this design is that learners are not restricted to only learn according to their learning styles. However, if there are learning resources that match their preferences, those resources are presented to them first. The validation of the educational outcomes of applying learning styles is out of the scope of this paper. The FSLS model, which consists of 4 categories, groups learners into the following [20]:

*   sensing or intuitive learners, based on how information is perceived;
*   visual or verbal learners, based on information-reception;
*   active or reflective learners, based on how information is processed;
*   sequential or global learners, based on how information is understood.

The preferred learning style of a learner can be determined through the Index of Learning Style questionnaire [24] which is made up of 44 questions (11 for each category), or by monitoring the behaviour and pattern of learners when they are offered free choices between different forms of learning content. To tackle the downside of the amount of time taken to answer all questions at once, the questionnaire was divided into respective categories, so a learner has to answer questions pertaining only to the relevant category in each course.

### 3.2. The Learner Model

The learner model is the data structure that defines a learner at any given time. It stores information that is useful in characterizing the learner and tracking the progress of the learner. The learner model for WASPEC is a hierarchical, multi-layered model, which is an overlay of the domain.

The following information is stored in the learner model:

*   General information;
*   Learning behaviour;
*   Learning state.

Learning behaviour details the preferences of the learner obtained through tasks or monitoring the learner's interactions with the learning system. The learning state details the progress of learning (level of knowledge, grades, and completed competencies) for a particular learner. This information, which is organised in a hierarchy and differs according to the relevant parameters in a course, is useful in creating relationships to the domain and semantic reasoning.

### 3.3. The Domain Model

Each course is organised as a networked hierarchy of concepts and competencies, defined at the inception of the course by the course instructor. Each concept can be represented by several competencies, with each competency organised into different levels on a

difficulty scale, corresponding to the difficulty element (5.8) of the Learning Object Metadata (LOM) standard [25]. The LOM standard, a data model proposed by IEEE's Learning Technology Standards Committee (LTSC), is one of the most widely used metadata standards for describing and manipulating learning objects. It also encourages reusability and sharing across various learning platforms. The standard includes more than 70 descriptors organised into nine categories, with each descriptor defining a specific aspect that can be considered in the description of an LO. Table 1 shows the relationship between concepts, courses, and competencies in a course for English as a language. Table 2 describes the relationship between element 5.8 of the LOM standard and user's level of knowledge.

Learning objects have been described differently, but for this study, a learning object (LO) is a reusable digital resource with a specific learning goal that can be used to support learning [5]. Figure 1 shows the relationships between concepts, competencies, and learning objects. A course can be represented by many concepts, and concepts can be cross-referenced in several courses. For each concept, there are several competencies, with each aiming to achieve a specific learning objective. Each competency is to be instructed by one or more LOs, with each attaining different learning objectives according to a subset of Bloom's Taxonomy of Learning [26].

**Table 1.** The hierarchical structure between concepts and competencies in a course.

| Difficulty | Competencies | Concept | Related Concepts |
|---|---|---|---|
| easy | Word order in Questions | | |
| medium | Subject and Object Questions | | |
| medium | Questions with prepositions | Questions | Prepositions |
| difficult | Negative questions | | |
| difficult | Indirect questions | | |

As an example, the competency 'Negative questions', in Table 1, can have three LOs which represent knowledge, comprehension, and application. Learning objects at the knowledge level correspond to resources a learner can listen to or read, which helps learners to remember the concept. Comprehension LOs further explore the concepts through examples, which furthers a better understanding of the concept. Application LOs correspond to exercises or real-world situations for the concept.

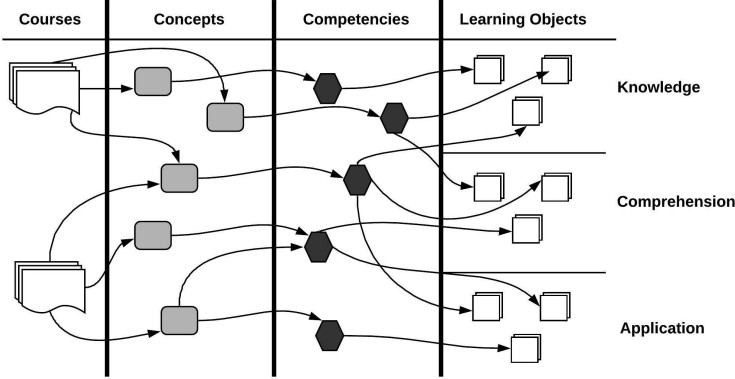

**Figure 1.** Structure of the domain model.

To define learning objects for each course, the learning resources are semantically annotated with elements of the LOM standard. Moodle has a variety of activity modules (learning resources) that can be used for both individual and collaborative learning, surveys, assessment, etc. To ensure interoperability and portability, the annotation of these modules in the LMS is described in the ontology. The mapping between personalisation parameters

and elements of the LOM standard is described in Table 2. The specific data elements included in this design are determined by the set of personalisation parameters chosen.

### 3.4. The Semantic Model for Waspec

The ontology developed for WASPEC enables knowledge modeling (which involves semantic mapping of the LMS relational database and WASPEC Service Framework relational database), semantic annotation of learning resources, and semantic reasoning. The classes, object properties, and data properties defined in the ontology are a representation of relevant content and components on the LMS platform, which includes information on the learner and domain model. It also includes the relationships between learning objects, metadata, and personalisation parameters.

The main classes of the ontology include Assessment, Cohort, Competency, Concept, Course, DataElement, Grade, Group, LearningObject, LOMetadata, Parameter, Questionnaire, User, UserPreference, and UserActivity. The basic organisation of a course is described by the Course class. Each course, represented by competencies and concepts, has learning objects that are used for instruction. The LOMetadata class describes the property of each learning object with the elements of the DataElement class, which is subsequently mapped to the Parameter class. The User class describes users on the platform. The Group class defines features of users which are course-specific, while the Cohort class defines features that are platform-wide.

For semantic reasoning with dynamic information, the UserActivity class stores information regarding the activities of the learners and their interactions with the system. The Grade class models the grades of the learners. These classes are vital for providing adaptivity in real-time during the learning process.

**Table 2.** Relationship between Personalisation Parameters and elements of the LOM standard in WASPEC.

| Parameter | Dimension | Nr. | Element | Metadata Value |
|---|---|---|---|---|
| Language Preference | English Romanian | 1.3 | Language | en ro |
| Felder-Silverman LS | global global global sequential sequential | 1.7 | Structure | atomic collection networked hierarchical networked |
| Felder-Silverman LS | active reflective | 5.1 | Interactivity Type | active expositive |
| Felder-Silverman LS | visual visual visual visual visual verbal verbal | 5.2 | Learning Resource Type | diagram figure graph simulation video lecture narrative text |
| Media Preference | text/image text/image text/image text/image audio video illustration | 5.2 | Learning Resource Type | diagram figure graph narrative text lecture video simulation |
| Motivational Level | low medium high | 5.4 | Semantic Density | low medium high |
| Level of Knowledge | beginner intermediate advanced | 5.8 | Difficulty | easy medium difficult |
| Felder-Silverman LS | sensing intuitive | 9.1 | Information Perception | facts/details theories/principles |

*3.5. The Architecture of Waspec*

Figure 2 describes the architecture of WASPEC, which is highly modular, allowing for interoperability with different LMSs. It is also flexible, which permits efficient modification and scalability. Personalisation on the platform is achieved by extending the LMS with web services, plugins, and adding semantic components. The WASPEC learning platform comprises three major high-level components: an LMS, a service framework, and a semantic framework.

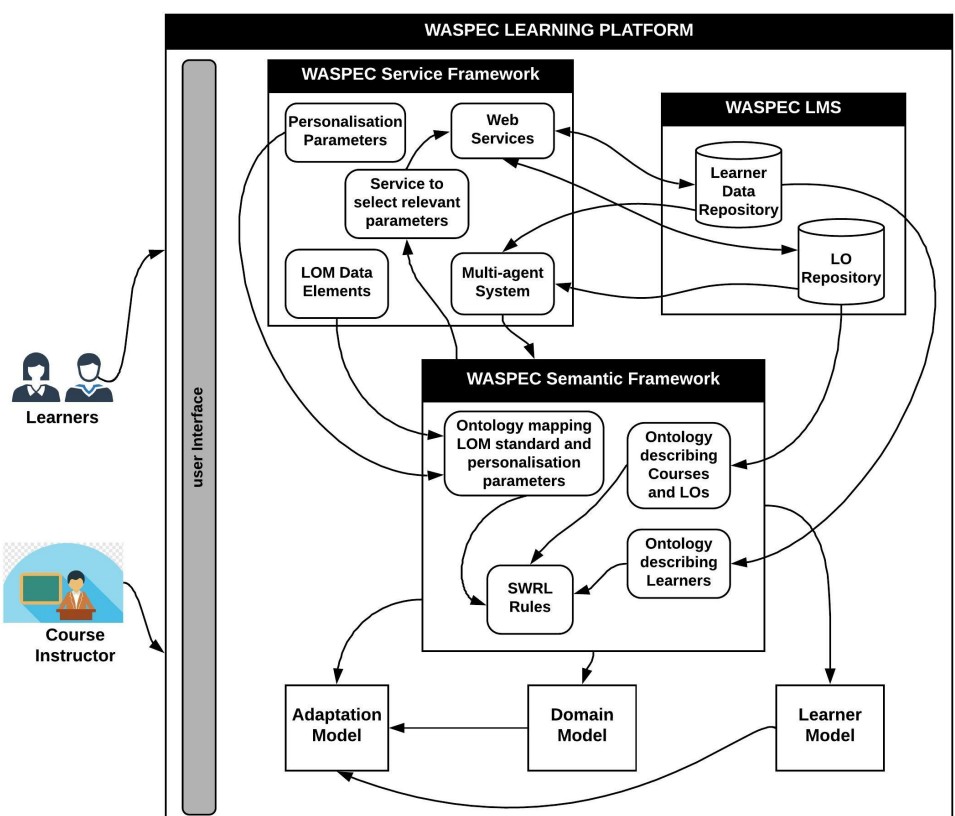

**Figure 2.** Architecture of WASPEC.

The learners and course instructor access the platform through a user interface, with only the course instructor privy to the service framework. The learners interact with the platform through the LMS, which stores information about their preferences and learning state, thus constituting the learner model. The course instructor adds courses and learning resources on the LMS platform. The service framework interacts with the LMS through web services. The application of web service technologies provides robust solutions for interoperability when multiple applications have to communicate with each other [10]. In this context, a service, which is a distant function that can be executed when called, is independent of implementation. This provides developers with the option to only call the services when needed, allowing for efficient integration of personalisation technologies. The web services are necessary for the semantic annotation of courses, learning resources, and learners on the LMS. The set of possible parameters and the data elements, they are related to, are also defined in the service framework.

The DBs of the service framework and the LMS are described as graph databases in the semantic framework. Information from these graphs, represented in the ontology, is used in the determination of relevant personalisation parameters for each course in the service framework. Semantic components defined in the semantic framework update the learner model with static and dynamic information of the learner's preferences, learning state, and learning behaviour. These graph databases are subsequently linked together in the creation of semantic rules and used in personalisation. The results of personalisation from the semantic framework are communicated to the LMS through web services.

### 3.6. The Process of Personalisation

As stated previously, there are two sets of possible parameters: the first set, which determines what the learners learn and the second set, which determines how the learners learn. These two sets of parameters are used in two distinct levels of personalisation.

### 3.6.1. Level 1 Personalisation

At this level of personalisation, LOs are categorised based on their difficulties according to the LOM standard specification. Learners are also categorised into different groups according to their knowledge levels. The groups include beginner, intermediate, and advanced as described in Table 2.

Thus, the features of a specific learner, at this level, can be expressed as:

$$F_U = < (lok, w) >, \tag{1}$$

where *lok* represents the level of knowledge of the learner, and *w* represents the linguistic value, which can be beginner, intermediate, or advanced.

An LO of any type, at this level, can be expressed as:

$$F_{LO} = < (d, v) >, \tag{2}$$

where *d* represents the difficulty of the *LO*, and *v* represents the metadata value, which can be easy, medium, or difficult.

The mapping between users and learning objects is executed by semantic rules which will be described in the implementation. Learners with higher levels of knowledge have access to learning objects below their levels, while learners with lower levels of knowledge can only access higher-level learning objects after completing competency requirements.

### 3.6.2. Level 2 Personalisation: A Multi-Parameter Personalisation Approach

To create learning paths from a set of multiple parameters for the second level of personalisation, their dimensions are combined in a Cartesian Product A × B (if all values are to be represented). For instance, if we consider a course that will be personalised with Media Preference (text_image, audio, video, illustration) and the Active/Reflective dimension of the FSLS model, we get:

$[text\_image, audio, video, illustration] \times [active, reflective]$,

This gives the possible set of learning paths represented below:

$\{\{text\_image, active\}, \{text\_image, reflective\}, \{audio, active\}, \{audio, reflective\},$
$\{video, active\}, \{video, reflective\}, \{illustration, active\}, \{illustration, reflective\}\}$

Thus, for this combination, there are eight (8) possible learning paths. When more parameters are combined, as the current state of the literature recommends, the number of possible learning paths increases rapidly. This reaffirms the need for a procedure of selecting parameters that are most relevant in a course (as they vary from one course to another).

Two metrics are used for recommending relevant parameters for personalisation at this level. The first is the Learning Object Representation based on Dimensions of a Personalisation parameter for all competencies (LOR-PD), and the second is the Complementary Ratio of Learning Objects based on Dimensions of a Personalisation parameter for each competency (CRLO-PD).

LOR-PD (for a personalisation parameter) is calculated by finding the quotient of the number of cells that are represented by learning objects and the total number of cells for each personalisation parameter. CRLO-PD (for a competency) is determined by getting the quotient of dimensions represented by learning objects and the total number of dimensions. The values of LOR-PD and CRLO-PD (derived from Table 3) are shown in Tables 4 and 5, respectively.

**Table 3.** Use Case Mapping between Competencies and dimensions of Personalisation Parameters.

| | Personalisation Parameters | | | | | |
| | Media Preference | | | | Active/Reflective FSLS | |
| **Competencies** | text/image | video | audio | illustration | active | reflective |
| Comp1 | $LO_1$ | $LO_4$ | $LO_5$ | | $LO_8$ | |
| Comp2 | | $LO_5$ | $LO_7$ | | $LO_9$ | $LO_{10}$ |
| Comp3 | $LO_2$ | | | | | $LO_{11}$ |
| Comp4 | $LO_3$ | | | | | |

**Table 4.** LOR-PD values for Personalisation Parameters.

| Personalisation Parameters | LOR-PD Values |
| --- | --- |
| Media Preference | 0.44 |
| Active/Reflective FSLS | 0.5 |

**Table 5.** CRLO-PD values for competencies.

| Competencies | Media Preference | Active/Reflective FSLS |
| --- | --- | --- |
| Comp1 | 0.75 | 0.5 |
| Comp2 | 0.5 | 1 |
| Comp3 | 0.25 | 0.5 |
| Comp4 | 0.25 | 0 |

The closer LOR-PD values are to 1, the more the dimensions of the parameter are represented with LOs, suggesting that such a parameter is suitable for personalisation in that course. The closer CRLO-PD values are to 1, the more the dimensions of the selected personalisation parameter are represented with LOs, suggesting that the competency can be fully personalised in all dimensions for that parameter. Active/Reflective FSLS, for instance, is fully represented for Comp2, but partially represented for Comp1 and Comp3. Thus, all learners will receive $LO_8$ and $LO_{11}$ (Table 3) for Comp2 and Comp3 respectively, but active learners will receive $LO_9$ and reflective learners will receive $LO_{10}$ for Comp2 if Active/Reflective FSLS is selected for personalisation.

These metrics are useful in the following ways:

- LOR-PD assists the course instructor in choosing parameters, mostly represented by learning objects, which are suitable for personalisation;
- LOR-PD and CRLO-PD specify competencies that require improvement for divergent dimensions of a selected parameter;
- For parameters that have more than two divergent dimensions (such as media preference), by analyzing LOR-PD values, a particular dimension can be eliminated. For example, 'illustration' in Media Preference in Table 3, can be eliminated because there are no LOs representing it for all competencies.
- CRLO-PD specifies when (and when not) to apply personalisation for each competency for selected parameters.

### 3.6.3. Selecting Relevant Parameters with Lor-Pd Values

One of the challenges of applying multiple parameters is the amount of time taken for learners to complete tasks/questionnaires to categorise them into different groups before the actual learning process commences. This is commonly referred to as the 'cold start' problem [27] in computing. For such systems that have to provide personalisation based on user information, at the start of the process, there is no information because the users have no interaction with the system. However, if the time and process taken to collect user information for personalisation is long and cumbersome, users may lack motivation and

drop out before the actual activity begins. Thus, any activity to categorise learners based on some criteria has a time factor attributed to it. The less time it takes to complete such an activity, the earlier learners can begin the learning process.

Selecting the relevant parameters based on LOR-PD values can be represented as an optimisation problem similar to the knapsack problem [28]. A personalisation parameter, $p$, takes a certain amount of time, $t$, (in minutes) to achieve satisfaction, $s$, which is represented by its LOR-PD value. The problem is defined as achieving the highest satisfaction in the least possible time, given an optimal time limit of $t_o$.

$N \in \Re$, and for all $1 \leq i \leq N$, $s_i, t_i \in \Re$. $S$ and $T$ represent the satisfaction and time sequences, where $S = (s_i)_{i=1}^N$ and $T = (t_i)_{i=1}^N$. If $P$ represents the sequence of parameters, $P = (p_i)_{i=1}^N$, we can define

$$P_s = (sp_i)_{i=1}^N \tag{3}$$

as a descending sequence of: $s_i \times (t_i \div t_o)$, for all $1 \leq i \leq N$.

The problem can be computationally solved using dynamic programming [29]. From the sequence of $P_s$, the course instructor can select a set of parameters, $x$, where $x \leq N$, which are suitable for personalisation for a course.

From Equation (2), the features of a LO, at this level, can thus be expressed as:

$$F_{LO} = < (d, v), (sp_1, w_1), (sp_2, w_2), \dots (sp_x, w_x) > . \tag{4}$$

From Equation (1), a learner, at this level, can thus be expressed as:

$$F_U = < (lok, q), (sp_1, w_1), (sp_2, w_2), \dots (sp_x, w_x) >, \tag{5}$$

where $sp$ represents the selected personalisation parameter (mapped to the metadata attributes) of the LO, and $w$ represents the linguistic term.

### 3.6.4. The Learning Process

Figure 3 shows the activity diagram for a learner on the WASPEC learning platform. The initial visit of a learner to the platform requires the student to register and complete general information. At this point, the user is not required to complete any questionnaire or task required for personalisation. When the user signs up for a course, for the first level of personalisation, the learner's knowledge level of the concepts represented in the course and the competencies required to complete the course is obtained; the learner model is then updated with this course-specific information.

For the second level of personalisation, the student is required to complete tasks or answer questionnaires that represent criteria for the most relevant personalisation parameters of that course. This information, which isn't course-specific, is then updated in the learner model and can be used in the personalisation of other courses. The student can subsequently explore learning content that is tailored to his/her preferences. Personalisation at this level is performed by adaptive navigation (link hiding, specifically).

The dynamic update of the learner model is based on determining the students' behaviour during the learning process, by analysing their interactions with the learning platform. This data which is vital for personalisation is managed by semantic modules which have been designed for their collection.

For monitoring and updating the learner model, three indexes will be used. The first index is the average grade of the course. This is obtained by computing the mean score of learning objects that assess the knowledge of the learner. If the learner's performance is satisfactory and above average, modifying the preferences of such a user in the learner model will not be necessary. However, if the average grade of the learner is subpar, the interaction between the learner and learning object will determine the other two indexes. For a below-average performance, the system changes the presentation of learning content to include learning objects outside the student's preferences. The time spent on different

learning objects serves as a second index. The third index is the ratio of the number of learning objects visited the total number of learning objects in the course.

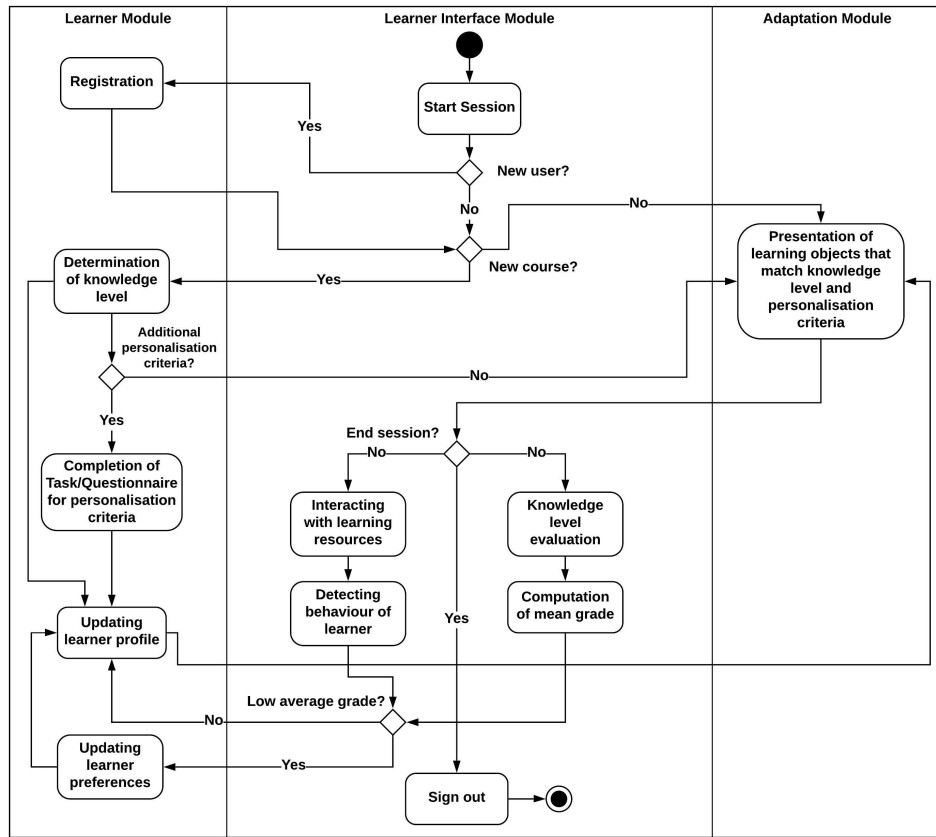

**Figure 3.** The learning process for learners in WASPEC.

These indexes provide the data that will be required to alter the learner model of the student for below-average performance. If the performance is subsequently improved by the additional presentation of learning content, the system can update the learner model to reflect those changes. Further presentation of learning content will be according to the new changes in the learner model.

## 4. Waspec: Model Implementation

This section describes the technical implementation of WASPEC's design. As stated in Section 3, WASPEC was developed by extending Moodle. The technical components will be described according to the modular components of the architecture in Figure 2.

### 4.1. Waspec Service Framework

The WASPEC service framework was implemented with Laravel [30], a PHP open-source web framework, which provided a visual platform to consume the services from Moodle. The platform enabled modular and independent metadata annotation to courses and their contents. Personalisation parameters and data elements of the LOM standard are also specified here. This approach ensures easy interoperability with other LMSs.

#### 4.1.1. Web Services

Most LMSs, such as Moodle, provide application programming interfaces (APIs) that provide easy transfer of data to and from the platform. Moodle provides web services that utilise several protocols. Simple Object Access Protocol (SOAP) was used in development because of its ability to support WSDL (Web Service Description Language) specification,

which makes discovery and integration with web services straightforward. To interact with Moodle, several web service functions were utilised.

Moodle, however, has a general approach in storing information on users and course content. To add extra metadata, other web services were implemented and added to Moodle's web service functions. The implemented web service functions enabled competencies and concepts to be added to courses, additional features to be added to learners, competencies to be assigned to learning resources, and learning resources to be enriched with metadata using the LOM elements.

### 4.1.2. Waspec Multi-Agent System

The multi-agent system for WASPEC was developed in SPADE (Smart Python multi-Agent Development Environment) [31], which is a MAS platform developed in Python, based on XMPP/Jabber technology. SPADE facilitates the design of a network of agents that can interact with each other through a communication protocol like FIPA-ACL (Foundation for Intelligent Physical Agents—Agent Communication Language) [32]. The agents developed for the WASPEC platform include:

- Ontology Update Agent;
- Learner Assessment Agent;
- Learner Performance Agent.

The Ontology Update Agent monitors specific tables of the LMS database and updates the ontology if there are any changes. The Learner Assessment Agent monitors each learner's progress during the learning process. If a learner has completed all assessment items for a particular level of knowledge, a message is sent to the Learner Performance Agent. The Learner Performance Agent monitors the performance of each learner for all levels level of knowledge. If a learner's performance is satisfactory, the learner moves to the next level. If a learner's performance is unsatisfactory, the learner's preferences can be changed according to the indexes specified in Section 3.

### *4.2. Waspec Semantic Framework*

The semantic framework for WASPEC includes Moodle's mapped database, the ontology describing the relationships between personalisation parameters and elements of the LOM standard, and a set of SWRL (Semantic Web Rule Language) rules.

### 4.2.1. Mapping Rdbs to Rdf Schemas

To semantically exploit Moodle's robust database system, the RDB was mapped to an RDF schema, to run SPARQL queries and enable semantic reasoning. Mapping Moodle's RDB to an RDF schema was done with the D2QR platform [33]. The D2RQ Platform allows RDBs to be accessed as virtual, read-only RDF graphs, without having to recreate the RDB into an RDF store. The main components of the D2RQ platform used in this design include a declarative mapping language that expresses relationships between an ontology and a relational data model, and a D2R server that provides an HTTP connection and a SPARQL Protocol endpoint over the database which can be accessed with a SPARQL client. With these components, the following functionalities are possible:

- Querying Moodle's database (which is non-RDF) using SPARQL;
- Accessing the contents of the RDBs (service framework and Moodle) as linked data over a network;
- Creating custom dumps of Moodle's database in RDF formats to load into an RDF store;
- Accessing information from the relational databases with a SPARQL client.

### 4.2.2. Creating Waspec Ontology

The first step in designing an ontology would be carefully specifying the goal and range of the ontology. Ontologies, which are thoroughly designed, can reliably describe a domain. However, it is important to balance the expression of the ontology design with

complexity. For this design, existing standard vocabularies and ontologies were exploited when mapping using the D2RQ platform. IEEE LOM standard was used for modeling learning objects. The resulting ontology visualised with Protégé is shown in Figure 4.

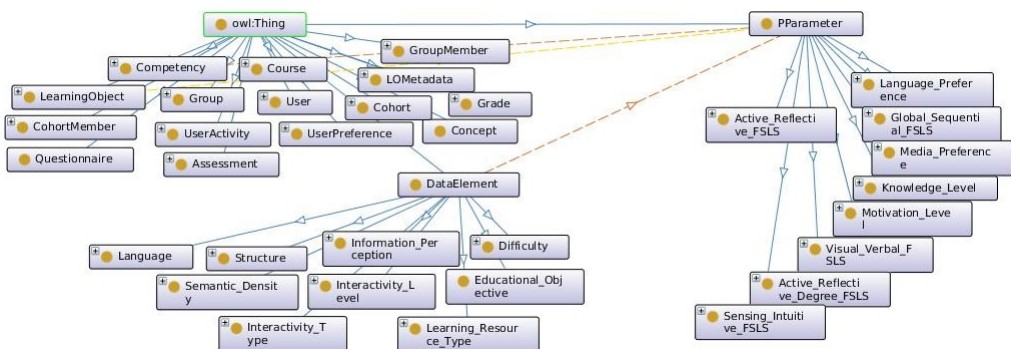

**Figure 4.** The ontology for WASPEC Semantic Framework.

The current literature presents three main approaches when accessing ontologies through a programming language [34]. The first approach uses a query language such as SPARQL. The drawbacks with query languages include the fact that it is not object-oriented and it is data-oriented. Being data-oriented, queries can be performed, but inferences cannot be made. The second approach utilises an API, such as the combination between (Web Ontology Language) OWL API and Jena (an open-source Semantic Web framework for Java). The third approach takes advantage of the similarities between object models and ontologies by using ontology-oriented programming.

This approach, using OWLready2 [34] (which is a python module that allows access to OWL ontologies) and Python, was used in the development of the ontology for WASPEC where classes, properties, and individuals in the ontology correspond to classes, attributes, and instances, respectively, in object models. It also allows for the definition of classes and hierarchies, variables and restrictions, the relationships between classes.

### 4.2.3. SWRL Rules

SWRL integrates rules, concepts, and the relationships between concepts defined in Web Ontology Language (OWL), thereby extending their expressiveness. Creating new knowledge, with SWRL, is accomplished by defining rule sets, which fundamentally serves as an inference engine [35]. SWRL rules, which are essentially 'if ..., then ...' associations, are executed in Pellet [36], a W3C (World Wide Web Consortium) reasoner, embedded in OWLready2.

## 5. Evaluation Method

Evaluating this system involved the visual presentation of personalizing a course to instruct the English Language to secondary school students according to the Common European Framework of Reference for Languages (CEFR) curriculum on the WASPEC Moodle platform. The modified course consisted of 4 concepts, 33 competencies (of different difficulties), and 64 learning objects. The LOs were annotated with metadata according to Table 2.

Figure 5 shows the resulting LOR-PD values for each personalisation parameter with regards to the time taken for testing on the service framework. From these values, the Visual/Verbal FSLS and Active/Reflective FSLS were selected as relevant parameters for personalisation for this course because all dimensions are represented and they have decent satisfaction values.

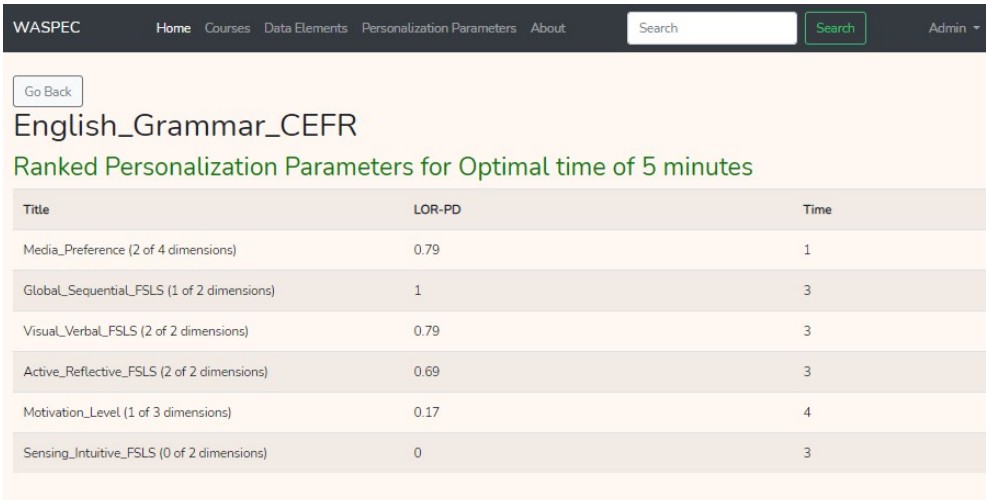

**Figure 5.** Personalisation parameters with LOR-PD indexes ranked with time (in minutes).

Hence, the learning resources on the course were personalised based on the level of knowledge (beginner, intermediate, and advanced for the first level of personalisation) and Visual/Verbal FSLS and Active/Reflective FSLS at the second level of personalisation. Figure 6 shows the resulting personalisation of learning resources on the Moodle platform when viewed with administrative privileges.

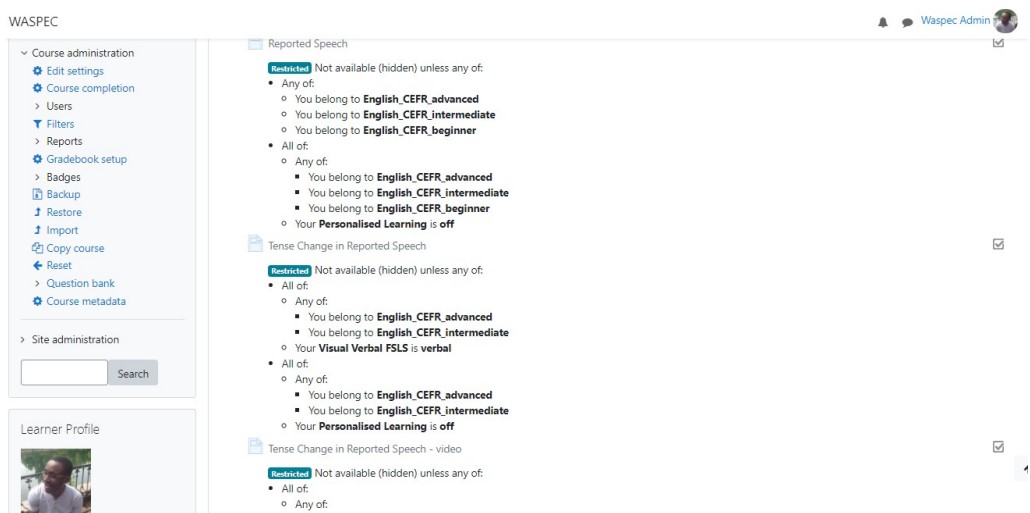

**Figure 6.** Personalisation viewed as an administrator.

Personalisation at the second level is dependent on CRLO-PD indexes as shown in Figure 7. The first LO (Prepositions of time) is personalised with two parameters because the CRL0-PD values for both parameters that the LO represents are 1. This means that other LOs are representing other divergent dimensions for each of the two parameters. For the second LO, Distinguishing between 'during', 'for', and 'while', only Visual/Verbal FSLS has a CRLO-PD value of 1; hence, it is the only parameter used for personalisation for that competency. For the third LO, neither parameters have a CRLO-PD value of 1 for the competency it represents; therefore, it is shown to all users. With this approach, all users are certain to receive learning resources for all competencies specified in the course, even though it does not match their preference at the second level of personalisation.

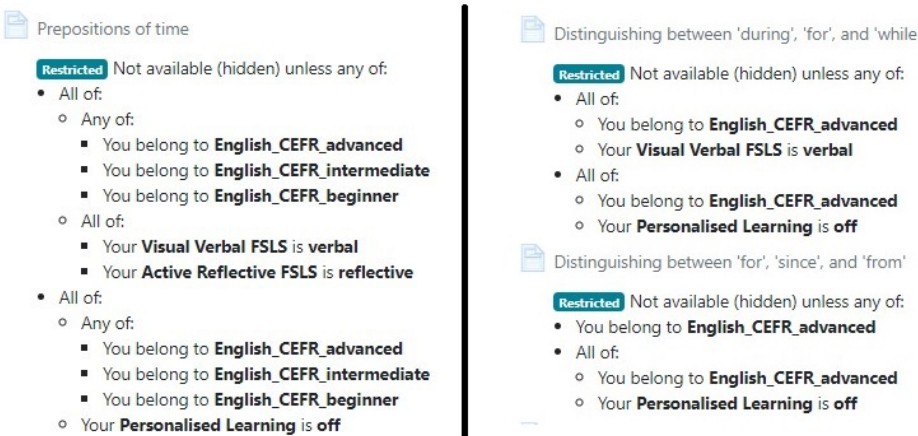

**Figure 7.** Application of CRLO-PD index at the second level of personalisation.

Figures 8 and 9 show how the course modules are viewed from a learner's perspective. Personalisation is completely hidden from the learner.

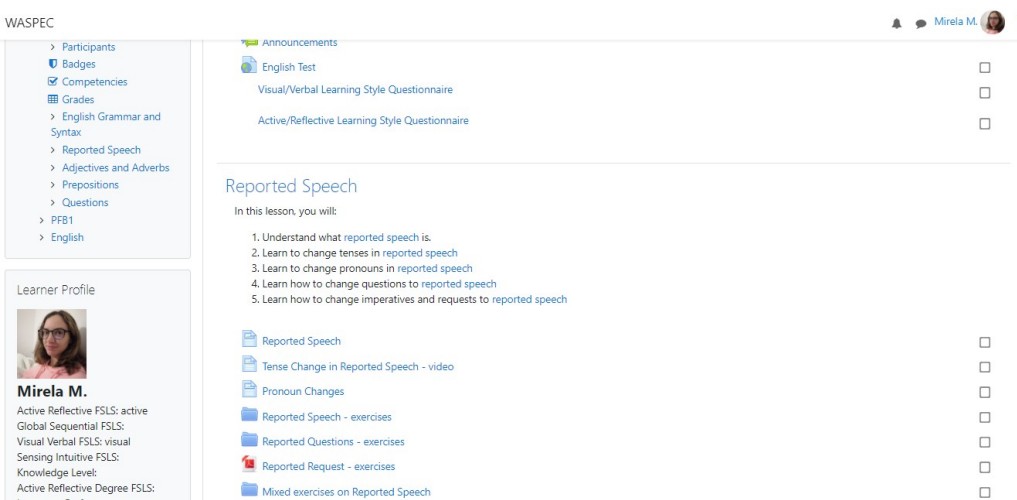

**Figure 8.** Course modules viewed as learners.

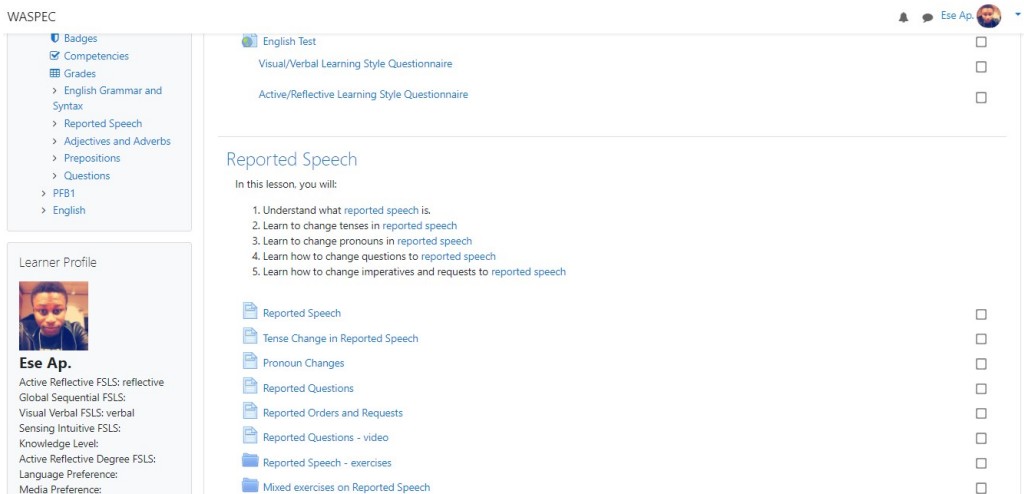

**Figure 9.** Course modules viewed as learners.

### 5.1. Participants and Evaluation Tool

Nineteen participants were involved in the process of validating WASPEC. These participants consisted of research students, course instructors, and professors familiar

with current trends in e-learning. Participants were required to respond to a Technology Acceptance Model (TAM) questionnaire. TAM is widely used in information science as a means of evaluating new designs [37]. The model covers several variables for evaluation, however, four sections were selected for our evaluation process.

### 5.2. Evaluation

The selected sections include Perceived Usefulness, Perceived Ease of Use, Perceived Intention to Use, and Perceived Attitude Towards Using. Participants had to respond by selecting options that range from 'strongly agree' to 'strongly disagree'.

#### 5.2.1. Perceived Usefulness

- Using the WASPEC platform increases productivity in selecting multiple parameters for personalisation with regards to available learning resources.
- Using the WASPEC platform to create multi-parameter personalised courses in a curriculum would increase teaching performance.
- Using the WASPEC platform would increase the effectiveness in creating personalised courses with multiple parameters.
- I find the WASPEC platform useful in creating personalised curricula.

#### 5.2.2. Perceived Ease of Use

- Learning to work with the WASPEC platform would be easy for me.
- I would find it easy to use the WASPEC platform to create multi-parameter personalised courses.
- It would be easy to become skillful in the use of the WASPEC platform.
- I would find the WASPEC platform easy and straightforward to use.

#### 5.2.3. Perceived Intention to Use

- If I had access to the WASPEC platform, I intend to adopt it.
- I would tend to frequently use the WASPEC platform in creating personalised courses if access is available.

#### 5.2.4. Perceived Attitude Towards Using

- Using the WASPEC platform in creating personalised learning environments has a positive influence.
- Using the WASPEC platform is an innovative approach towards personalisation.

## 6. Results

With the values of 'strongly agree' to 'strongly disagree' ranging from 1 to 5, the mean and median values of the responses of the participants are calculated. A mean or median value close to 1 indicates satisfaction with the WASPEC platform. On the other hand, mean or median values close to 5 suggest dissatisfaction with the platform. The results of the responses are displayed in Table 6, showing the different sections of the TAM questionnaire. For more details, please refer to the Supplementary Material.

**Table 6.** The mean and median values from the evaluation of the WASPEC platform.

|        | Usefulness | Ease of Use | Intention to Use | Attitude Towards Using |
|--------|------------|-------------|------------------|------------------------|
| Mean   | 1.87       | 2.08        | 2.11             | 1.82                   |
| Median | 2.00       | 2.00        | 2.00             | 2.00                   |

## 7. Discussion and Comparison of Waspec to Related Work

From Table 6, we can see an acceptable level of satisfaction from the participants who evaluated the WASPEC platform. The mean values ranged between 1 and 2 and are far

from 5. For instance, the mean value of the attitude of participants towards the platform was 1.82 and perceived usefulness was 1.87.

In Table 7, comparisons between WASPEC and related designs are made, based on the following criteria:

- Personalisation criteria: The systems will be evaluated based on the (number of) criteria included for personalisation.
- Standardisation approach: The standards of metadata annotation of learning resources (if used) will be evaluated here.
- Personalisation approach This will discuss how each design achieves personalisation of learning resources or instruction.

**Table 7.** Comparing related works based on personalisation criteria, standardisation, and personalisation approach.

| | **Personalisation Criteria** | **Standardisation** | **Personalisation Approach** |
|---|---|---|---|
| WASPEC | A set of possible parameters, which includes learning style (FSLS model), level of knowledge, media preference, language preference, and motivational level | Learning objects are specified by metadata annotation using elements of the IEEE LOM standard | Rule-based reasoning with semantic ontologies and web services |
| [12] | Learning Styles-VAK model and global/sequential class of FSLS model | None specified | Rule-based approach which matches learning resources to different dimensions of learning styles |
| [13] | FSLS model dimensions and level of knowledge | Metadata annotation was done with IEEE LOM and Dublin Core | Rule-based reasoning with semantic ontologies and web services |
| [14] | Dimensions of the FSLS model | None specified | Case-based reasoning of clustering students based on data mining results, and subsequently using inferences to provide personalisation for similar situations |
| [15] | Level of knowledge and learning style (FSLS model) | None specified | Case-based reasoning of clustering based on collected data. |
| [16] | Learner's background and learning objectives | None specified | Artificial Intelligence Mapping |
| [17] | Dimensions of the FSLS model | None specified | Rule-based reasoning |
| [18] | User needs and context | SCORM specification | Rule-based reasoning |

### 8. Conclusions

This paper has presented the design and implementation of an e-learning system, WASPEC, which delivers a curriculum of personalised courses to learners using Moodle as an LMS. The model and design of this system address the questions of combining multiple parameters in a personalisation strategy. This was achieved by defining two metrics, LOR-PD and CRLO-PD which aid in determining the relevant parameters that can be used for personalisation based on the learning resources in a course. It also approaches personalisation by merging semantic web functionalities to the Moodle platform. The WASPEC platform was created by extending Moodle through web services, plugins, and ontologies.

Although Moodle was chosen as an LMS, for this design, because of its wide use and robust storage system and capabilities for extendibility, this approach is highly modular

and can be interoperable with other LMSs. This also allows for efficiency and ease of improvements and scalability.

Future work will involve extending the ontology to incorporate other personalisation parameters and extended evaluation with learners.

**Supplementary Materials:** The following are available at https://www.mdpi.com/article/10.3390/computers10050059/s1.

**Funding:** This research received no external funding and The APC for this paper was funded by Faculty of Computer Science, Alexandru Ioan Cuza

**Institutional Review Board Statement:** Not applicable.

**Informed Consent Statement:** Informed consent was obtained from all the participants that responded to the questionnaire.

**Data Availability Statement:** The data presented in this study are available on request from the corresponding author.

**Conflicts of Interest:** The authors declare no conflict of interest.

## Abbreviations

The following abbreviations are used in this manuscript:

| | |
|---|---|
| MOODLE | Modular Object-Oriented Dynamic Learning environment |
| WASPEC | Weighted Agent System For Personalised E-learning Curricula |
| LMS | Learning Management System |
| RDF | Resource Description Framework |
| XML | Extensible Markup Language |
| MAS | Multi-Agent System |
| ITS | Intelligent Tutoring System |
| VLE | Virtual Learning Environment |
| FSLS | Felder-Silverman Learning Style |
| LOM | Learning Object Metadata |
| LO | Learning Object |
| IEEE | Institute of Electrical and Electronics Engineers |
| API | Application Programming Interfaces |
| SOAP | Simple Object Access Protocol |
| WSDL | Web Service Description Language |
| SWRL | Semantic Web Rule Language |
| OWL | Web Ontology Language |

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
