# Peer review of "The Design of WASPEC: A Fully Personalised Moodle System Using Semantic Web Technologies"

_computers, doi:10.3390/computers10050059_

Round 1
Reviewer 1 Report
The idea is promising, it has potential, but there are aspects that clearly need to be considered for publication. In particular, I detect one relevant aspect to improve (related work) and another that is absolutely essential (evaluation of the improvement in academic performance between two groups, one control and the other in which the educational aid agent is applied).
Regarding the first point, a much more in-depth SLR would be desirable, indicating (and justifying) the review methodology used. These existing elements should be explained in detail, and their performance compared with the authors' proposal should be compared in an evaluation section that is clearly insufficient.
In the second aspect, a complete evaluation by means of an experiment is key, detailing the course in which it is applied, the characterization of the sample, and a statistically relevant justification of the improvement in academic results between the target group and the control group.
Author Response
- Related work has been extended to include other designs
- The result section has been extended to include evaluation by different participants
Reviewer 2 Report
This is a good academic work that needs to be improved in order to be of real value for readers.
Major issues:
1.- The Learner Model used is based on FOAF. There have been several efforts to deliver standards for Learner Model, like IEEE PAPI (not very succesful) or the proposal made in the IMS Consortium (IMS LIP), mainly related to accesibility but not only restricted to them or the standardization groups created at ISO. Previous relevant works based on the ontological model of the learner were, very rightly, based on standardization models coming from the e-learning field.
See for example:
Zine, O., Derouich, A., & Talbi, A. (2019). IMS Compliant Ontological Learner Model for Adaptive E-Learning Environments. International Journal of Emerging Technologies in Learning, 14(16).
The authors need to justify the use of FOAF without considering any of the characteristics modeled in education-specific person models.
2.- Authors claim their architecture to be interoperable with other LMSs, in addition to Moodle. This needs to be justified.
First of all, it is not clear from the paper how your approach can be developed in a 'regular' Moodle instance, I assume as a Moodle plugin. This needs to be described.
Secondly, related work on interoperable e-learning architectures and mechanisms to communicate learning tools should be briefly approached in the paper, e.g. introduction. See for example:
Anido-Rifón, L., et al. "A step ahead in E-learning Standardization: Building learning systems from reusable and interoperable software components." WW2002, The Eleventh International World Wide Web Conference. Honolulu, Hawaii, USA. 2002.
3.- Section 5 on results needs to be elaborated. As it is, this work is kept at a theoretical level. It is not clear how the authors have evaluated its benefits beyond a theoretical comparison to related work. Have you used it in practice with real students? Do you get it tested in an experts focus group?
4.- Related work is much broader than what is presented in section 6. Proposals for pesonalized learning can be found, for instance at:
Caputi, V., & Garrido, A. (2015). Student-oriented planning of e-learning contents for Moodle. Journal of Network and Computer Applications, 53, 115-127.
Additional related work, for example those above, should be considered in section 4
Other issues:
1.- The introduction needs to provide a more elaborated insight into personalization in e-learning. I suggest to introduce, LMSs, ITS and the efforts made even at the level of standardization bodies (see below)
2.- I do not see section 2 as a proper literature review. This section provides an overview on the techniques to be used in this work. I would include the contents of this sections within the Introduction.
3.- The 'cold start' problem, when there is not enough information about the learner interaction at the beginning should be better explained
4.- LOM provides over 60 elements. There are other standards for describing learning objects, for example ISO has its own proposal MLR. The selection of those elements chosen needs to be justified.
Author Response
- In the original design, FOAF was used. However, for now, FOAF will not be used. In subsequent improvements, other ontological learner models for e-learning environments will be considered.
- Personalisation is done mostly on the service framework. The results of personalisation are transferred to Moodle through web-services. Several web-services were also implemented to enable metadata annotation of users and course modules on Moodle.
- The results section have been extended to present evaluation of the system.
- Related work has also been extended to include other similar and designs.
- The introduction section has been extended to include some important technologies
- The previous literature view has been merged with the introduction. It was replaced with related work.
- I don't quite understand the 'cold start' problem you're referring to. If you could elaborate, I'd be grateful.
Reviewer 3 Report
Multi-level personalization in e-learning systems is an important topic. The manuscript presents a well-structured design with detailed description of the proposed solution.
One of the key system personalization parameters is student learning styles. The underlying logic is that students have distinct learning styles and if they learn accordingly, they will have better results. However, there is strong evidence from neuroscience that suggest the opposite. In fact, learning styles are one of the most prevalent neuromyths in education (Dekker, Lee, Howard-Jones, & Jolles, 2012; Newton, 2015; Newton & Miah, 2017; Rato, Abreu, & Castro-Caldas, 2013; Torrijos-Muelas, González-Víllora, & Bodoque-Osma, 2021). Therefore, authors need to study the literature and address this issue by justifying their choices.
Also, the Results section is too short. It needs to be enriched significantly with results from the actual implementation of the system, based on a concrete evaluation method.
Dekker, S., Lee, N., Howard-Jones, P., & Jolles, J. (2012). Neuromyths in Education: Prevalence and Predictors of Misconceptions among Teachers . Frontiers in Psychology . Retrieved from https://www.frontiersin.org/article/10.3389/fpsyg.2012.00429
Newton, P. M. (2015). The Learning Styles Myth is Thriving in Higher Education. Frontiers in Psychology, 6, 1908. https://doi.org/10.3389/fpsyg.2015.01908
Newton, P. M., & Miah, M. (2017). Evidence-Based Higher Education – Is the Learning Styles ‘Myth’ Important? Frontiers in Psychology, 8, 444. https://doi.org/10.3389/fpsyg.2017.00444
Rato, J. R., Abreu, A. M., & Castro-Caldas, A. (2013). Neuromyths in education: what is fact and what is fiction for Portuguese teachers? Educational Research, 55(4), 441–453. https://doi.org/10.1080/00131881.2013.844947
Torrijos-Muelas, M., González-Víllora, S., & Bodoque-Osma, A. R. (2021). The Persistence of Neuromyths in the Educational Settings: A Systematic Review. Frontiers in Psychology, 11, 591923. https://doi.org/10.3389/fpsyg.2020.591923
Author Response
- The results section has been extended to include evaluation by different participants.
- Regarding learning styles, this paper is not necessarily focused on validating the logic that absolutely matching teaching strategy with learning styles will produce better results (as that is out of the scope of this research). The proposal here is that students have preferences, and in a course with different learning materials, if learning resources matching student preferences are presented to student first, it might increase motivation to learn. However, the student is not limited to a certain style, and if performance is below average, the student will learn alternatively. Also, in cases where there are no learning resources, matching a student's learning style, the general course material is presented to the student as specified by the CRLO-PD index. Please see the third paragraph of page 6 for justification.
Round 2
Reviewer 1 Report
The authors have done a good job of revising the article and have improved on many of the issues raised. There are still some points, in the related work and validation, that could be improved, but the quality has increased significantly.
Author Response
The validation process and result section has been improved.
Reviewer 3 Report
The revised manuscript addresses the issues raised. I suggest inserting a new section 5 “Methods” which should be separated from the next section 6 “Results”.
Author Response
A different section has been added. "Evaluation Method" which is separate from the "Results" section.